# Functional Analysis of Alkaline Phosphatase in Whitefly *Bemisia tabaci* (Middle East Asia Minor 1 and Mediterranean) on Different Host Plants

**DOI:** 10.3390/genes12040497

**Published:** 2021-03-29

**Authors:** Wen-Hao Han, Chi Zou, Li-Xin Qian, Chao Wang, Xiao-Wei Wang, Yin-Quan Liu, Xin-Ru Wang

**Affiliations:** Ministry of Agriculture Key Laboratory of Molecular Biology of Crop Pathogens and Insects, Institute of Insect Sciences, Zhejiang University, Hangzhou 310058, China; wenhao_han@zju.edu.cn (W.-H.H.); zouchichi@126.com (C.Z.); lxqian@zju.edu.cn (L.-X.Q.); chwang2016@zju.edu.cn (C.W.); xwwang@zju.edu.cn (X.-W.W.); yqliu@zju.edu.cn (Y.-Q.L.)

**Keywords:** ALPs, *Bemisia tabaci*, comparative genomics, functional genomics

## Abstract

Alkaline phosphatases (ALPs: EC 3.1.3.1) are ubiquitous enzymes and play crucial roles in the fundamental phosphate uptake and secretory processes. Although insects are regarded as the most diverse group of organisms, the current understanding of ALP roles in insects is limited. As one type of destructive agricultural pest, whitefly *Bemisia tabaci*, a phloem feeder and invasive species, can cause extensive crop damage through feeding and transmitting plant diseases. In this study, we retrieved five ALP genes in MEAM1 whitefly, nine ALP genes in MED whitefly via comparative genomics approaches. Compared with nine other insects, whiteflies’ ALP gene family members did not undergo significant expansion during insect evolution, and whiteflies’ ALP genes were dispersed. Moreover, whiteflies’ ALP gene family was conserved among insects and emerged before speciation via phylogenetic analysis. Whiteflies’ ALP gene expression profiles presented that most ALP genes have different expression patterns after feeding on cotton or tobacco plants. Female/male MED whiteflies possessed higher ALP activities on both cotton and tobacco plants irrespective of sex, relative to MEAM1 whiteflies. Meanwhile, adult MED whiteflies possessed higher ALP activity in both whole insect and salivary samples, relative to MEAM1 whiteflies. We also found that both MED and MEAM1 whiteflies could upregulate ALP activities after feeding on cotton compared with feeding on tobacco plants. These findings demonstrated the functions of whiteflies ALPs and will assist the further study of the genomic evolution of insect ALPs.

## 1. Introduction

Alkaline phosphatases (ALPs: EC 3.1.3.1) are plasma membrane-bound glycoproteins and are extensively distributed from prokaryotes to higher eukaryotes [1]. As crucial components and abundant enzymes in most organisms, ALPs have conserved and typical features, such as mediating hydrolase/transferase reactions, having a dimeric structure, containing Zn^2+^/Mg^2+^ metal ions as essential active sites for enzymatic activity, and regulating subunit interaction [2]. Compared with prokaryotes, the specific characteristics in mammalian ALPs include different tissue-specific forms, higher specific activity, lower heat stability, and more alkaline pH optimum [3]. Unlike mammalian ALPs that have received broad attention, studies on insect ALPs are still perplexing and long overdue. Since the first report of an insect ALP identified in the malpighian tunes of *Bombyx mori* in 1957 [4], most literature focuses on model insects or some species with “importance,” such as *Drosophila melanogaster*, *Apis mellifera*, *Manduca sexta*, and *Bombyx mori*, etc. Previous studies revealed that insect ALPs show tissue-specific distributions with membrane transport, such as intestinal epithelial cells, malpighian tubules, haemolymph, and salivary glands [5,6,7,8]. Besides the function in dephosphorylating compounds, insect ALPs also participate in the modulation of insect developmental hormones, juvenile hormone (JH), and 20-hydroxyecdysone (20E) [9,10,11,12]. Recently, an “unexpected” role of insect ALPs received more attention, particularly in regulating insecticide resistance—a paradigm from a brown planthopper, in which soluble ALP possesses a strong response to hormone and insecticide [13]. Another case is the tobacco hornworm, in which ALP could manipulate reactions in *Bacillus thuringiensis* Cry1Ab toxin [8]. Obviously, regulation of enzyme activities would be a useful tool for insects during their lifecycle, especially in adapting to environment stresses.

The whitefly *Bemisia tabaci* (Gennadius) (Aleyrodidae: Hemiptera) is a species complex with more than 40 cryptic species that threaten crop protection via direct feeding and transmitting plant diseases [14,15]. Two invasive whiteflies, MEAM1 (formerly biotype B) and MED (formerly Q), tend to be more aggressive than other cryptic species, thereby rapidly replacing native whiteflies. In China, tracing MEAM1 whiteflies’ invasive history, the first case was reported in the mid-1990s, while MED whiteflies were first detected on ornamental plants in Yunnan province in 2003 [16]. However, MED whiteflies dominated over MEAM1 and even replaced some native whiteflies in a short period, then spread across the country in China [17,18]. Previous studies attempted to explain this phenomenon via comparison of the different characteristics between MEAM1 and MED whiteflies, including feeding and reproductive behavior, host plant range, insecticide resistance, virus transmission, and endosymbiont species [19,20,21,22], yet little is known regarding the underlying mechanism explaining why the MED whitefly is so aggressive and how it replaces other cryptic species. A possible explanation for this issue is that whiteflies’ displacement is contributed to by multiple factors, such as hosts, environments, specific characteristics among different whitefly cryptic species, etc. Here, we are particularly interested in host plants’ adaptation to whitefly invasion. Considering the diversity of insects’ ALP functions, we speculate that the distinct ALPs between MED and MEAM1 whiteflies might be a factor for the MED species to possess higher host plant adaptability than other cryptic species.

Funk first identified whitefly ALPs in salivary glands and secreted saliva in 2001 [23], but the current understanding of whitefly ALP functions is merely the tip of the iceberg. Only two recent studies separately revealed host plants’ effects on ALP activity and ALP activity in the different development stages of whiteflies [24,25]. Interestingly, previous studies also revealed that some secondary plant substrates could regulate insect ALP activity in rice leaffolder [26,27]. Therefore, investigating the interactions between the host plant and insect ALPs would provide fundamental information to this field. Significant progress in whole genome sequencing technology empowers a comprehensive investigation of whitefly ALPs [28,29]. This study utilized comparative genomics approaches to predict, identify, and analyze ALP gene family members in MEAM1 and MED whitefly genomes. By comparing ALP gene expression patterns and ALP activities, we found that MED whiteflies show relatively higher ALP activity relative to MEAM1 whiteflies. Moreover, both invasive whiteflies showed higher ALP activities when on cotton than on tobacco. Our study investigates the functions of whiteflies ALPs in host plant adaptation and provides valuable insights into invasive insect species replacement of native species.

## 2. Materials and Methods

### 2.1. Data Set Collections and B. tabaci ALP Sequences Annotations

The recently published whitefly genomes of MEAM1 and MED were downloaded from Whitefly Genome Database (http://www.whiteflygenomics.org/cgi-bin/bta/index.cgi (accessed on 31 January 2018) [29] and Giga Database (http://gigadb.org/dataset/100286 accessed on 31 January 2018)) [28], respectively. The information and sequences of ALP proteins of *Aedes aegypti*, *Drosophila melanogaster*, *Bombyx mori*, *Heliothis virescens*, *Tribolium castaneum*, *Nilaparvata lugens*, *Acyrthosiphon pisum*, *Nasonia vitripennis*, and *Apis mellifera* (Accession numbers listed in Appendix A) were downloaded from InsectBase (http://www.insect-genome.com/ accessed on 31 January 2018) [30] and NCBI and subsequently used as queries to search against both whitefly genomes through standalone BlastP and tBlastN programs with a stringent E value cut-off (≤10^−20^). All non-redundant hits were subjected to the NCBI Conserved Domain Database (http://www.ncbi.nlm.nih.gov/cdd accessed on 6 March 2018) [31] and SMART (http://smart.embl-heidelberg.de/ 6 March 2018) [32] for further confirmation and redundant ALP proteins filtrations.

The ExPASy Compute pI/Mw tool (http://ca.expasy.org/tools/pi_tool.html accessed on 9 April 2018) [33] and CELLO v2.5 server (http://cello.life.nctu.edu.tw/ accessed on 21 April 2018) [34] were, respectively, utilized to predict the physicochemical properties and subcellular localization predictions of whiteflies’ full-length ALP proteins. In addition, the signal peptides of whiteflies’ ALP proteins were predicted by the SignalP 5.0 server (http://www.cbs.dtu.dk/services/SignalP/ accessed on 2 May 2018) [35]. Glycosylphosphatidylinositol (GPI)-anchoring sites were predicted by GPI modification site prediction servers (PredGPI: http://gpcr2.biocomp.unibo.it/gpipe/ accessed on 9 May 2018) [36].

### 2.2. Phylogenetic Analysis, Conserved Domain Prediction, and dN/dS Values for Orthologous ALP Pairs Calculation

Multiple sequence alignments of all ALP proteins were conducted by Clustal Omega [37] with default parameters. Phylogenetic trees were constructed using IQ-tree [38,39] with the maximum likelihood (ML) method. The parameters were as follows—corrected Akaike Information Criterion and Bayesian Information Criterion with LG + R5 and WAG + G4 models, 1000 bootstrap replicates. All phylogenetic trees were viewed and edited by iTol (https://itol.embl.de/ accessed on 18 May 2018) [32]. Here, we named whiteflies’ ALP genes according to the previous study [40,41]. For example, ALPs in MEAM1 and MED whiteflies begin with the initials of the genus and species names, ‘Bt,’ followed by ‘ALP’ and subsequently by a number indicating the homologous relationship with *D. melanogaster*. The neighbor-joining (NJ) phylogenetic trees of ALP proteins were constructed using MEGA7 [42] with 1000 bootstrap replicates to confirm the reliability of ML trees further.

The conserved protein domains of ALP proteins from MEAM1 and MED whiteflies were assessed via Motif Elicitation (MEME, http://meme-suite.org/tools/meme accessed on 26 May 2018) [43]. TBtools [44] was used to draw and modify those motifs in each BtALP protein. Genes from different species present on the same branch of the phylogenetic tree were designated as orthologs [45]. The ALP orthologs from MEAM1 and MED whiteflies were aligned and submitted to PAL2NAL [46] to estimate the nonsynonymous substitutions rate (dN) and synonymous substitution rate (dS). The ratio of dN/dS was calculated to assess the selection pressure of orthologs.

### 2.3. Insect Rearing

In this study, the cryptic whitefly species MEAM1 and MED were reared on healthy cotton plants (*Gossypium hirsutum* L. cv. Zhemian 1793), then separately transferred into healthy cotton or healthy tobacco plants (*Nicotiana tabacum* cv. NC89) for 10 generations in insect-proof cages kept at 26 ± 1 °C with a 16:8 light:dark cycle. Cotton or tobacco seeds were sown into pots and cultivated in isolation until they had at least five real leaves before exposure to whiteflies.

### 2.4. RNA Isolation and Real-Time Quantitative Reverse Transcription PCR (qRT-PCR)

Total RNA was extracted from the whiteflies using TRIzol (Invitrogen, 15596026, Carlsbad, CA, USA). The quantity and quality of RNA were evaluated using a NanoDrop2000 Spectrophotometer (NanoDrop Technologies, Wilmington, DE, USA). Complementary DNA (cDNA) was synthesized using the SYBR PrimeScript reverse transcription-PCR (RT-PCR) kit II (Takara, RR037A, Dalian, China). qPCR was performed on CFX96^TM^ Real-Time system (Bio-Rad, Foster City, CA, USA) with SYBR green detection (Takara, RR071A). All protocols were according to the manufacturer’s instructions. *Elongation factor-1a* (*EF-1a*) and *β-actin* (*ACTB*) are reference genes used for ALP gene expression profiles. Primers used in this study are listed in Appendix A.

### 2.5. ALP Activity Assays

For single female/male whitefly fed on cotton/tobacco, the ALP activity was analyzed according to the previous study [23,24]. One single whitefly was homogenized in 100 μL grinding buffer (10 mM MgCl2, 150 mM NaCl, 0.1% Nonidet P-40, 1 mM PMSF, 50 mM Tris, pH 7.5), then was centrifuged in 12,000× *g* for 10 min at 4 °C. The supernatant as the ALP enzyme solution was subjected to assay. The reaction was conducted in a 96-well plate and incubated at 37 °C for 30 min, including 10 μL ALP enzyme solution, 30 μL p-nitrophenyl Phosphate solution (0.75 mM pNPP), and 170 μL N-tris (hydroxymethyl) methyl-3-aminopropanesulfonic acid (100 mM TAPS, pH 7.8). Absorbance readings were taken at 405 nm on a varioskan flash multimode reader (Thermo Scientific, Waltham, MA, USA). The ALP enzyme unit (OD/min × mL) represented absorbance change of 1 for the reaction per minute and ml of enzyme solution.

For whole insects’ ALP activity assays, a pool of mixed sex adult whiteflies (100) fed on cotton/tobacco were homogenized in 500 μL grinding buffer, respectively. The following methods were performed according to the above-described protocol in single female/male whitefly ALP activity assays.

For whitefly saliva ALP activity assays, approximately 400 mixed-sex adult MEAM1 and MED whiteflies fed on cotton/tobacco were transferred into a feeding chamber, respectively. Saliva collection and the appreciate pH for TAPS buffer were identified as previously described [23]. The reaction was conducted in 100 μL saliva solution, 30 μL 0.75 mM pNPP, and 170 μL 100 mM TAPS (pH 8.0). The measurement protocols followed the above-described method in single female/male whitefly ALP activity assays.

### 2.6. Statistical Methods

The relative expression levels of each gene were calculated by the comparative CT method (2^−ΔΔCt^). Statistical analysis was carried out by SPSS 20.0 using the data obtained from three separate cDNA sets of three independent biological replicates. Mev4.0 software was used to generate a heat map of the qPCR results. The student’s two-tailed t test was applied for the pool of female or male whiteflies, saliva ALP activities analysis. Differences indicated by * were judged significant when *p* < 0.05.

## 3. Results

### 3.1. Identification of ALP Gene Family Members in MEAM1 and MED Whiteflies

ALP protein sequences from *Aedes aegypti*, *Drosophila melanogaster*, *Bombyx mori* as queries were searched against MEAM1 and MED whitefly draft genomes [28,29]. Five and nine putative ALP genes were identified and further verified with the NCBI Conserved Domain Database and SMART. In parallel, the ALP genes were also identified in other insects among five orders—*Heliothis virescens*, *Tribolium castaneum*, *Nilaparvata lugens*, *Acyrthosiphon pisum*, *Nasonia vitripennis*, and *Apis mellifera* (Table 1). The lengths of the deduced amino acid sequences, molecular weights, and isoelectric points (pI) of BtALP genes were predicted in Table 2, suggesting distinct characteristics among MEAM1 and MED whitefly ALPs. Since a protein’s subcellular localization is highly correlated with its functions, we found that half of whitefly ALPs were predicted to be periplasmic proteins with signal peptides located in the N-terminal region and GPI-anchor sites located in the C-terminal region except for BtALP5-Q (Table 2).

### 3.2. Phylogenetic Analysis of MEAM1 and MED Whiteflies ALPs Compared with Other Insects

To investigate ALP proteins’ evolutionary relationships in MEAM1 and MED whiteflies and other insects surveyed here, an unrooted phylogenetic tree was constructed by IQ-tree (Figure 1). Meanwhile, a neighbor-joining tree was reconstructed by MEGA, supporting the reliability of the unrooted phylogenetic tree. ALP proteins in MEAM1 and MED whiteflies were classified into six groups, while BtALP5-B and BtALP9-Q were located on the furthest branches, perhaps reflecting that they might be more distantly related compared with other ALPs among Hemiptera species. Besides, most ALPs from Hemiptera species were clustered together and presented a closer relationship to Diptera species. Interestingly, most ALPs from Lepidoptera species were clustered tightly in one group, perhaps reflecting the conserved feature of Lepidoptera ALP evolution.

### 3.3. Conserved Protein Domain, Amino Acid Sequence Analysis of MEAM1 and MED Whiteflies ALPs

To investigate the orthologous relationships among ALP gene family members between MEAM1 and MED whiteflies, a separate maximum-likelihood phylogenetic tree was constructed using all the BtALP proteins (Figure 2A). The topology showed five pairs of orthologous ALP genes in MEAM1 and MED whiteflies, sharing high bootstrap values in the terminal branches. By analyzing the motif compositions, we found that all the ALP proteins in MEAM1 have corresponding orthologs in MED whiteflies. Interestingly, most ALP proteins contained six to nine conserved motifs, except for BtALP1-Q to BtALP3-Q with four motifs and BtALP7-Q with two motifs (Figure 2B and Appendix A). These results suggested that the orthologous proteins are possibly conserved in two cryptic species, while diverse structures of ALP proteins only found in MED whiteflies might imply distinct functions.

Furthermore, the dN/dS ratios of ALP orthologs in MEAM1 and MED were assessed (Table 3). The dN/dS ratio results were all under one, consistent with the previous study that found the average Ka/Ks ratio of MEAM1 and MED was 0.225 [47], suggesting ALP was under purifying selection during whitefly evolution.

To further explore the amino acid sequence conservation, all ALP proteins were aligned. Pairwise and multiple sequence alignment indicated that a signal peptide was present in the N-terminal region of most predicted periplasmic ALP proteins, except for BtALP1-Q. Besides, in those putative periplasmic ALP proteins with a signal peptide, a GPI-anchoring site was observed in the C-terminal regions as well, excluding BtALP2-Q. In other ALP proteins predicted as cytoplasmic or extracellular, no such type of signal peptides with GPI-anchoring sites were observed. The signal peptides and GPI-anchoring site prediction (Appendix A) indicated that all the ALP proteins in MEAM1 are membrane-bound ALPs excluding BtALP1-B, while in MED, only BtALP8-Q was a putative membrane-bound ALP protein.

### 3.4. Expression Profiles of ALP Genes in MEAM1 and MED Whiteflies Fed on Cotton and Tobacco Plants

A previous study demonstrated that ALP facilitates B whitefly feeding on different plant species compared to *Trialeurodes vaporariorum* during long feeding periods [24]. To investigate whether different host plants affect whitefly ALPs, we examined the expression patterns in *B. tabaci* MEAM1 and MED fed on different host plants, i.e., cotton and tobacco (Figure 3). When fed on cotton, the expression level of most ALP genes from MED whiteflies showed upregulation, except for BtALP1-Q, 2-Q, and 3-Q, compared to those of MEAM1 whiteflies. Only the BtALP5-Q of MED whiteflies exhibited a high expression level when fed on tobacco. Almost all MEAM1 whitefly ALP genes showed no significant change of expression levels on two different host plants. These results suggested that the diversity of *B. tabaci* expression patterns might indicate a MED whitefly-specific function in host plant adaptation.

### 3.5. ALPs Activities Analysis in MEAM1 and MED Whiteflies Fed on Cotton and Tobacco Plants

To investigate the direct role of host plants on whiteflies’ ALP activity, we examined the enzyme activity according to the previous description [24]. Whiteflies fed on cotton have higher ALPs activity than those fed on tobacco, regardless of the gender of MEAM1 and MED whiteflies. MED whiteflies presented higher ALP activity than MEAM1 whiteflies, both in the female and male (Figure 4). A previous study also indicated whitefly saliva contained ALP activity, and ALP aids whiteflies feeding on different host plants [23,24]. We further compared the ALP activity in whiteflies’ saliva with whole insects. MED whiteflies presented higher ALP activity than MEAM1 whiteflies, no matter whether in the saliva or whole insects. In addition, whiteflies fed on cotton have higher ALP activity in the saliva as well (Figure 5). These results strongly suggested that MED has increased ALP activity relative to MEAM1, consistent with the identified divergent family members of ALP proteins in the MED whitefly genome.

## 4. Discussion and Conclusions

Significant progress has been made in demonstrating insects’ ALP functions, such as actions in gut absorption and transportation, regulation of cuticle sclerotization, and mediation of neural and renal during development [5,7,48]. There is limited information regarding insects’ ALP comparative genomics and functional genomics analysis. In this study, we first identified two invasive whiteflies ALP gene family members through the genome and provided insight into insects’ ALP evolution. First, although MED and MEAM1 whiteflies shared almost the same size genome, an expanded ALP gene family exists in MED whiteflies (Figure 1). Interestingly, the insect’s ALP gene family size did not increase proportionally to the corresponding genome sizes in this study. For example, only two ALP genes were identified in *N. lugens*, even though it has a larger genome size relative to *B. tabaci*. We speculated that these insects are undergoing different evolution experiences, during which gene loss and gene gain events occurred randomly. Second, MED and MEAM1 whiteflies presented divergent characteristics in the ALP conserved domains. By analyzing signal peptides and GPI-anchoring sites, we found that most MEAM1 whiteflies’ ALPs are membrane-bound proteins, while MED whiteflies only carried one membrane-bound protein (Appendix A). Obviously, signal peptides and GPI-anchoring site prediction could provide a valuable clue to distinguish the soluble/membrane-bound forms among whitefly ALP gene family members. Regardless of MEAM1 and MED whiteflies’ ALPs, orthologues shared high similarity, and the extra four BtALP-Q genes presented diverse structures. Considering the tissue-specific functions of ALPs in insects, we speculated that these non-orthologous ALPs in MED whiteflies had unique functions. A paradigm from *B. mori*, two identified gut ALP proteins, a membrane-bound form (m-ALP) and a soluble form (s-ALP), located in a different region of the midgut, showed distinct differences in enzymatic activity and the structure of the sugar side chain [49,50,51,52]. Future studies on queries along this line, such as (1) to verify the soluble/membrane-bound form of ALPs in MED and MEAM1 whiteflies tissues, such as the gut, salivary glands, or reproductive systems, and (2) to investigate the function of specific non-orthologous ALPs in MED whiteflies, will be conducive to understanding the research of physiology, biochemistry, and genetics in insect ALPs.

ALP can hydrolyze orthophosphate monoesters, therefore activating absorption of metabolites and transport processes [2]. MED whiteflies showed strong resistance to the major classes of insecticide [53], suggesting that the larger ALP gene family size and more robust enzyme activity in MED might benefit its survival. By comparing pesticide applied to whiteflies’ ALP activities, it would demonstrate the possible mechanism of core enzymes underlying insect resistance to pesticides [8,13,54]. In addition, ALPs can manipulate the relative digestive enzymes to cope with different host plants for polyphagous insects. By comparing with whiteflies fed on tobacco plants, our results showed that, in whole insects or saliva, the MED whiteflies’ ALP activity showed significant upregulation (Figure 4 and Figure 5). Similar to the previous study, host plants’ suitability could shape MEAM1 and MED whiteflies’ competition, thereby triggering replacement. For instance, MEAM1 could displace MED on cabbage and tomato while being replaced by MED on pepper [22]. Another study also revealed that MEAM1 whiteflies possessed divergent ALP activity among different host plants (cotton, tomato, celery, and cabbage) and showed a superior ability to regulate ALP than *T. vaporariorum* [24]. Recently, MED whiteflies possessed more aggressive and higher adaptability compared with other cryptic species. Further research will investigate whether the ability of MED whiteflies to present higher ALP activity on different plants is a probable explanation for MED rapidly replacing MEAM1 and native species.

Considering that mammalian ALPs could regulate sucrose hydrolysis [2] and sucrose is the main ingredient in plant phloem juice that provides the nutrition for piercing and sucking insects, we are particularly interested in the relationships between whiteflies’ ALP activity and their “sucrose food”. Whiteflies’ saliva contains various enzymes that facilitate feeding on plants. Our results showed that ALP activities in MED and MEAM1 whiteflies’ saliva were remarkably higher on cotton plants, relative to tobacco plants (Figure 5). More interestingly, Funk also found that the whiteflies’ saliva ALP activity increased when sucrose concentration in the feeding solution increased [23]. So far, we do not know whether different sucrose concentration between the two plants is the key factor triggering ALP expression in whiteflies’ saliva. Besides, on either cotton or tobacco plants, MED whiteflies possessed relative higher ALP activity in the individual insect, or saliva, relative to MEAM1 whiteflies. Do sucrose or other ingredients from host plants decide the MED and MEAM1 whiteflies’ feeding behavior? Future research will address the above issues to reveal the host plants’ “role” in managing invasive whiteflies.

In this study, we first identified all ALP genes in MEAM1 and MED whiteflies. We analyzed the conserved gene structures, protein domain, dN/dS values of all ALP proteins. Phylogenetic analysis revealed that all the ALP proteins in whitefly are dispersed with the other insects’ ALPs. Expression patterns of ALP genes were distinct in MEAM1 or MED whiteflies after feeding on cotton or tobacco plants. ALP activities presented higher in MED whiteflies (adult female/male, whole insect, and saliva), relative to MEAM1 whiteflies. These findings contribute to a better understanding of the evolution of the insect ALPs family and provide possible clues for why invasive MED whiteflies rapidly dominated over MEAM1 in China.

## Figures and Tables

**Figure 1 genes-12-00497-f001:**
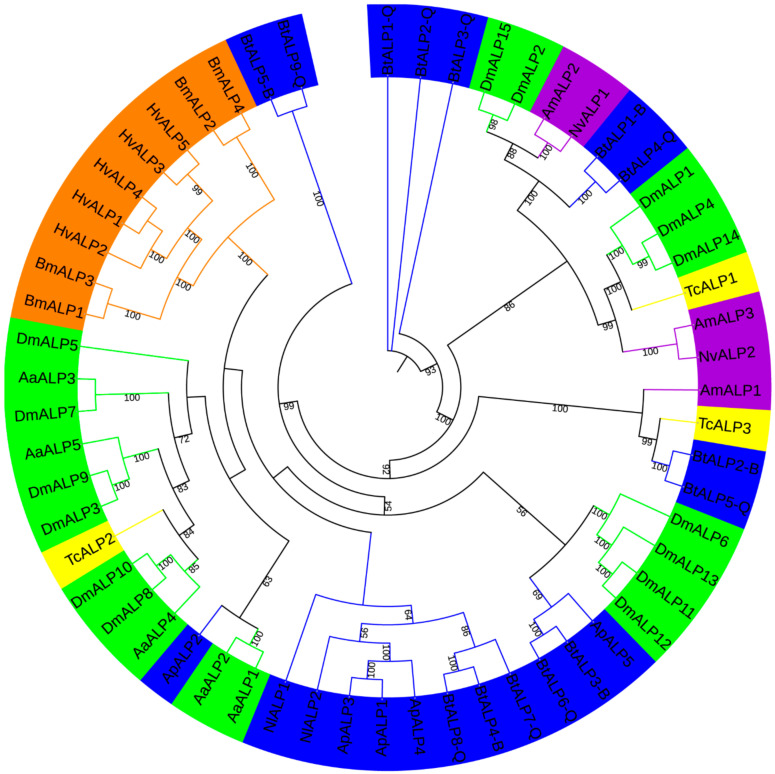
A maximum-likelihood phylogenetic tree generated by IQ-tree of ALP amino acid sequences from *B. tabaci* MEAM1 and MED and other insects. Bootstrap values are shown on each node, parameters used in generating this phylogenetic tree are shown in Methods. The insects from different orders are shown in different colors. The blue color represents Hemiptera (Nl, *Nilaparvata lugens*; Ap, *Acyrthosiphon pisum*; Bt, *Bemisia tabaci* MEAM1 and MED); the green color represents Diptera (Aa, *Aedes aegypti*; Dm, *Drosophila melanogaster*); the purple color represents Hymenoptera (Am, *Apis mellifera*; Nv, *Nasonia vitripennis*); the yellow color represents Coleoptera (Tc, *Tribolium castaneum*); and the orange color represents Lepidoptera (Bm, *Bombyx mori*; Hv, *Heliothis virescens*).

**Figure 2 genes-12-00497-f002:**
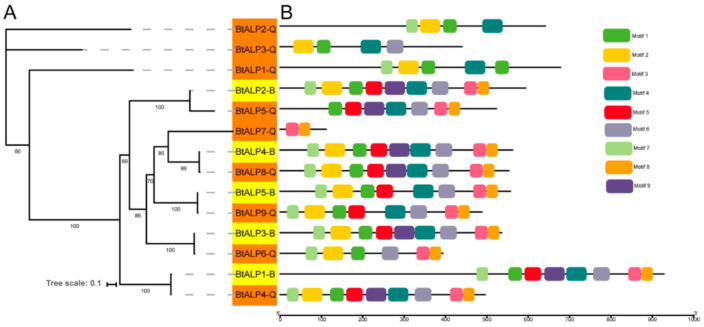
Phylogenetic relationships and protein motif analysis of *B. tabaci* MEAM1 and MED. (**A**)—the maximum-likelihood phylogenetic tree generated by IQ-tree and the bootstrap test was performed with 1000 replicates. Yellow shading marks ALPs in *B. tabaci* MEAM1. Orange shading marks ALPs in *B. tabaci* MED. (**B**)—all motifs were identified by the MEME database with the complete amino acid sequences of ALPs, and are presented in Appendix A.

**Figure 3 genes-12-00497-f003:**
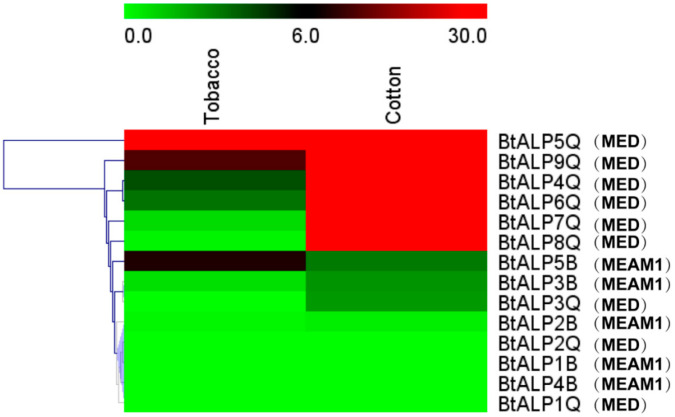
Analysis of expression profiles of ALP genes from *B. tabaci* MEAM1 and MED. Heat map showing the real-time quantitative PCR analysis results of BtALP genes in two plants. The bar colors vary from green (low) to red (high), representing the scale of relative expression levels. Each row represents a *B. tabaci* MEAM1 and MED gene.

**Figure 4 genes-12-00497-f004:**
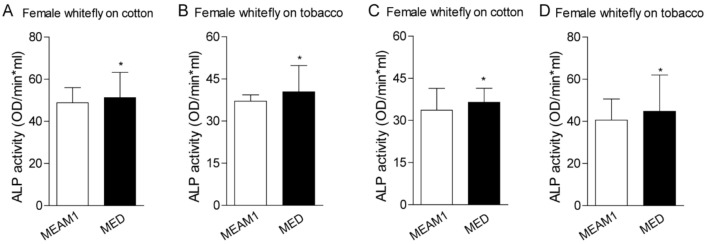
ALP activity analysis in whiteflies fed on cotton and tobacco. Each dot represents one female/male whitefly ALP activity. (**A**,**B**)—female whitefly on the cotton and tobacco from MEAM1 and MED whiteflies; (**C**,**D**)—male whitefly on the cotton and tobacco from MEAM1 and MED whiteflies. * indicates *p* < 0.05.

**Figure 5 genes-12-00497-f005:**
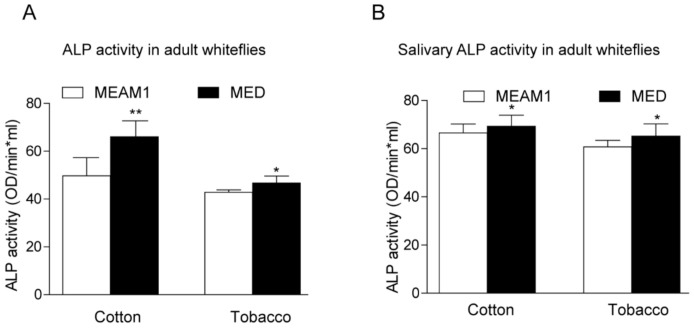
ALP activity analysis in whole insects (**A**) and whiteflies’ saliva (**B**) after feeding on different hosts. * indicates *p* < 0.05, ** *p* < 0.01.

**Table 1 genes-12-00497-t001:** Alkaline phosphatase (ALP) genes identified in *Bemisia tabaci* MEAM1 and MED and nine other insects.

Order	Species	ALP Gene Number
Coleoptera	*Tribolium castaneum*	3
Diptera	*Aedes aegypti*	5
	*Drosophila melanogaster*	15
Hemiptera	*Nilaparvata lugens*	2
	*Acyrthosiphon pisum*	5
	*Bemisia tabaci* MEAM1	5
	*Bemisia tabaci* MED	9
Hymenoptera	*Nasonia vitripennis*	2
	*Apis mellifera*	3
Lepidoptera	*Bombyx mori*	4
	*Heliothis virescens*	5

**Table 2 genes-12-00497-t002:** Summary of ALP genes found in *Bemisia tabaci* MEAM1 and MED.

Species Complex	Gene Name	Gene Identifier	CDS	AA	pI	Mw (kDa)	Predicted Subcellular Location	Signal Peptide
*Bemisia tabaci* MEAM1	*BtALP1-B*	Bta06445	2784	927	5.97	103.45	Cytoplasmic	_
	*BtALP2-B*	Bta07544	1782	593	8.09	64.795	Periplasmic	N 1–26
	*BtALP3-B*	Bta09616	1608	535	5.93	58.02	Periplasmic	N 1–22
	*BtALP4-B*	Bta14984	1686	561	5.83	61.157	Periplasmic	N 1–23
	*BtALP5-B*	Bta14395	1671	556	5.56	61.122	Extracellular	N 1–17
*Bemisia tabaci* MED	*BtALP1-Q*	BTA023249.1	2037	678	6.67	75.832	Periplasmic	_
	*BtALP2-Q*	BTA018369.1	1926	641	7.26	70.873	Periplasmic	N 1–23
	*BtALP3-Q*	BTA023373.1	1323	440	6.24	48.033	Extracellular	_
	*BtALP4-Q*	BTA003226.1	1491	496	6.26	54.505	Extracellular/Cytoplasmic	_
	*BtALP5-Q*	BTA006612.1	1572	523	6.64	56.565	Periplasmic	_
	*BtALP6-Q*	BTA027509.1	1185	394	5.27	42.659	Extracellular/Periplasmic	_
	*BtALP7-Q*	BTA017150.1	336	111	6.57	11.941	Cytoplasmic	_
	*BtALP8-Q*	BTA001657.1	1662	553	5.95	60.476	Periplasmic	N 1–26
	*BtALP9-Q*	BTA024811.1	1467	488	5.64	53.675	Extracellular	

**Table 3 genes-12-00497-t003:** The dN/dS ratio of ALP orthologs in whitefly.

Ortholog 1	Ortholog 2	dN	dS	dN/dS
*BtALP2-B*	*BtALP5-Q*	0.1349	0.3704	0.364
*BtALP4-B*	*BtALP8-Q*	0.0396	0.1236	0.321
*BtALP5-B*	*BtALP9-Q*	0.002	0.0195	0.1
*BtALP3-B*	*BtALP6-Q*	0.0158	0.0789	0.2
*BtALP1-B*	*BtALP4-Q*	0.0021	0.0342	0.061

## Data Availability

Data is contained within the article and Appendix A.

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
