# Peer review of "Functional Analysis of Alkaline Phosphatase in Whitefly Bemisia tabaci (Middle East Asia Minor 1 and Mediterranean) on Different Host Plants"

_genes, 2021, doi:10.3390/genes12040497_

Round 1

Reviewer 1 Report

P3, line126: Why 10 generations?

P11, line 279: The legend can be added " after cultured on different host". And the "on host plants" can be deleted in the figure.

P11, line 283: "little is known..." may change to "There is limited information..."

P11, lines288-290: Please provide any report to sport this speculation, or deleted!

P12, lines312-317: Please provide any evidence for ALP related to pesticide resistance. 

Reviewer 2 Report

This manuscript is a study of alkaline phosphatase genes in two whitefly populations. The authors phylogenetically compared ALP genes among insect species, compared predicted protein domains, and predicted secretion/localization of ALP proteins. They also analyzed the expression and activity of ALP in male/female whiteflies on different host plants. While this work can be a resource for the study of this gene class, substantial editing of the writing and clarification of the statistical analysis would strengthen the paper. Finally, additional controls are recommended for the activity assays to validate the conclusions described by the authors. 

Editorial Comments:
The English needs to be clarified so that the reader clearly understands what is being said. This must be done throughout the document, but here are some examples:
line 18: what does dispersed mean in this context? This could mean that the genes were dispersed among chromosomes. If so, they should be called genes rather than proteins. 
line 20: 'expressed patterns' should be 'expression patterns'
line 141: do you mean to say 'grinding'?

General Comments:
line 87: provide citations for the genomes
line 114: the naming convention for BtALP should be defined at first use.
154: it would be helpful to indicate at the start of this description that this is a salivary assay at the beginning of this paragraph.

It would be helpful for the introduction to describe host plant defense relevant to whitefly ALP activity. This would help the reader understand the choice to test ALP activity on different host plants. 

213-218 the english is particularly confusing, please clarify. 

Figure 2: The text supporting this figure could be clarified to more specifically describe the figure elements and the authors' interpretations of them. 

Figure 3 seems more appropriate for supplementary information. I encourage the authors to represent this in a more informative way if it remains in the main body of the manuscript. Minimally, the protein sequences are very hard to read. 

Figure 4: It is unclear which gene comes from MED versus MEAM1. It would be helpful to explicitly label these on the figure. In addition, the phylogeny aspect of the figure is not adequately explained in the text, and the legend should be more clearly labelled. 

Figure 5: The difference in ALP expression seems negligible from this figure. What does the astrix mean? The data points appear to have a wide distribution with a high degree of variability. I think the authors need to perform additional statistical analysis of these data and clarify their methods in the manuscript to convince readers that ALP expression is indeed up-regulated in MEAM1 and on cotton plants. 

Figure 6: How do we know that ALP activity is not higher in the host plant sap and therefore being detected in the insect after ingestion? The authors could provide a control showing ALP expression in host plant tissue. In addition, a gut clearing step would ensure that residual sap in the insects' guts is not responsible for ALP activity. 

Round 2

Reviewer 2 Report

The authors have addressed the comments and the manuscript is much improved in clarity and data presentation.